Whole-genome sequence analyses of Glaesserella parasuis isolates reveals extensive genomic variation and diverse antibiotic resistance determinants

Wan Xiulin 1
Li Xinhui 2
Osmundson Todd 3
Li Chunling lclclare@163.com 4
Yan He yanhe@scut.edu.cn 1
1 School of Food Science and Engineering, South China University of Technology , Guangzhou , China
2 Department of Microbiology, University of Wisconsin-La Crosse , La Crosse , United States of America
3 Department of Biology, University of Wisconsin-La Crosse , La Crosse , United States of America
4 Institute of Animal Health Guangdong Academy of Agricultural Sciences , Guangzhou , China
Gillespie Joseph
Electronic publication date: 2020 Jun 22
Publication date: 2020
Volume: 8
Electronic Location ID: e9293
Received 2019 Dec 24; Accepted 2020 May 13
Copyright: ©2020 Wan et al.
Copyright year: 2020
Copyright holder: Wan et al.
License: This is an open access article distributed under the terms of the Creative Commons Attribution License, which permits unrestricted use, distribution, reproduction and adaptation in any medium and for any purpose provided that it is properly attributed. For attribution, the original author(s), title, publication source (PeerJ) and either DOI or URL of the article must be cited.
License URL: https://creativecommons.org/licenses/by/4.0/

Keywords: Glaesserella parasuis, Mobile genetic elements, Phylogeny, Whole-genome sequencing, Antibiotic resistance genes

Funding: National Key Basic Research Program 2016YFD0500606 Natural Science Foundation of China 31772776 Key-Area Research and Development Program of Guangdong Province 2019B020217002 This work was supported by the National Key Basic Research Program (Grant No. 2016YFD0500606), the Natural Science Foundation of China (Grant No. 31772776), Key-Area Research and Development Program of Guangdong Province (Grant No. 2019B020217002). The funders had no role in study design, data collection and analysis, decision to publish, or preparation of the manuscript.

==============================
Background

Glaesserella parasuis (G. parasuis) is a respiratory pathogen of swine and the etiological agent of Glässer’s disease. The structural organization of genetic information, antibiotic resistance genes, potential pathogenicity, and evolutionary relationships among global G. parasuis strains remain unclear. The aim of this study was to better understand patterns of genetic variation, antibiotic resistance factors, and virulence mechanisms of this pathogen.

Methods

The whole-genome sequence of a ST328 isolate from diseased swine in China was determined using Pacbio RS II and Illumina MiSeq platforms and compared with 54 isolates from China sequenced in this study and 39 strains from China and eigtht other countries sequenced by previously. Patterns of genetic variation, antibiotic resistance, and virulence mechanisms were investigated in relation to the phylogeny of the isolates. Electrotransformation experiments were performed to confirm the ability of pYL1—a plasmid observed in ST328—to confer antibiotic resistance.

Results

The ST328 genome contained a novel Tn6678 transposon harbouring a unique resistance determinant. It also contained a small broad-host-range plasmid pYL1 carrying aac(6’)-Ie-aph(2”)-Ia and blaROB-1; when transferred to Staphylococcus aureus RN4220 by electroporation, this plasmid was highly stable under kanamycin selection. Most (85.13–91.74%) of the genetic variation between G. parasuis isolates was observed in the accessory genomes. Phylogenetic analysis revealed two major subgroups distinguished by country of origin, serotype, and multilocus sequence type (MLST). Novel virulence factors (gigP, malQ, and gmhA) and drug resistance genes (norA, bacA, ksgA, and bcr) in G. parasuis were identified. Resistance determinants (sul2, aph(3”)-Ib, norA, bacA, ksgA, and bcr) were widespread across isolates, regardless of serovar, isolation source, or geographical location.

Conclusions

Our comparative genomic analysis of worldwide G. parasuis isolates provides valuable insight into the emergence and transmission of G. parasuis in the swine industry. The result suggests the importance of transposon-related and/or plasmid-related gene variations in the evolution of G. parasuis.

Introduction

Glaesserella parasuis, a gram-negative bacterium in the family Pasteurellaceae (Dickerman, Bandara & Inzana, 2020), is a respiratory pathogen that affects swine. It is the etiological agent of Glässer’s disease, which can lead to pneumonia without signs of systemic disease (Brockmeier, 2004). As China is one of the world’s largest pork producers, with more than 463 million pigs accounting for approximately 50% of global population (Zhou et al., 2013), G. parasuis outbreaks in this country could pose a significant threat to pig health and economic loss worldwide (Brockmeier et al., 2014). Disease progression and severity are influenced by virulence and antibiotic resistance, both of which can result from evolutionary processes including mutation and horizontal gene transfer (Deng et al., 2019). Although antibiotic resistance may incur fitness costs in terms of virulence, the two phenomena may also act synergistically (Geisinger & Isberg, 2017).

Antimicrobial agents are widely used to prevent and control G. parasuis infection; however, overuse of antibiotics for non-therapeutic applications—including promoting growth in healthy individuals—has resulted in the evolution of antibiotic resistant G. parasuis in farming environments (Zhao et al., 2018). Antibiotic resistance in G. parasuis is mainly conferred by a combination of transferable antibiotic resistance genes (ARGs) and multiple target gene mutations. To date, two β-lactam resistance genes (blaROB−1 and blaTEM), an aminoglycoside-resistance gene (aac (6′)-Ib-cr) and a mutation in the six copies of the 23S rRNA gene, associated with macrolide resistance, have been reported in G. parasuis (Doi & Arakawa, 2007; San et al., 2007 ; Guo et al., 2012). G. parasuis strains often harbour multiple resistance genes and multi-drug resistance phenotypes, thus deterring clinical treatment.

PCR-based studies of G. parasuis strains have identified ARGs including tetB, aph(3′)-Ib, aph(6)-Id, floR, sul1, and sul2 (Wissing, Nicolet & Boerlin, 2001; San et al., 2007; Zhao et al., 2018), and virulence factors including the haemolysin operon (hhdBA), iron acquisition genes (cirA, tbpA/B and fhuA), the restriction modification system hsdS, and genes involved in sialic acid utilization (neuraminidase nanH and sialyltransferase genes neuA, siaB and lsgB) (Martinez-Moliner et al., 2012; Costa-Hurtado & Aragon, 2013). Recently, whole-genome sequencing (WGS) has emerged as a powerful tool for predicting antibiotic resistance and pathogenic potential in G. parasuis. For instance, Li et al. (2013) reported two G. parasuis strains with potential resistance towards the antibiotics ciprofloxacin, trimethoprim, and penicillin, based on the presence of associated resistance genes; Nicholson et al. (2018) reported genomic differences in the toxin-antitoxin systems between phenotypically distinct G. parasuis strains from Japan and Sweden; and Bello-Orti et al. (2014) noted the role of mobile genetic elements and strain-specific accessory genes in fostering high genomic diversity between pathogenic strains of the same serovar from diseased pigs in Japan, China, and the USA.

Though significant effort has been focused on exploring ARGs, virulence factors and other genetic characteristics of various G. parasuis strains, the structural organization of genetic information, ARGs, potential pathogenicity determinants, and evolutionary relationships among global G. parasuis strains remain unclear. In this study, we sequenced a multidrug-resistant isolate from diseased swine in Dongguan, China, then compared this genome sequence with those of 54 isolates from China sequenced by us and 39 strains from China and eight other countries sequenced by other researchers in order to improve our understanding of genomic diversity in G. parasuis and provide information for gaining better control to treat these infections.

Materials & Methods

Isolates

The multidrug-resistant G. parasuis isolate HPS-1 examined in this study belongs to serotype 4 and was originally isolated from the lungs of a pig suffering from Glässer’s disease in a commercial pig farm in Dongguan city, Guangdong province, China, in 2017. Susceptibility to 19 antimicrobial agents was determined by the disc agar diffusion method and the broth microdilution method (Pruller et al., 2017). The isolate was determined to be resistant to β-lactams, aminoglycosides, macrolides, quinolones, lincomycin, and sulfonamides (Table S1).

The other 54 G. parasuis isolates were obtained from diseased pigs from more than 20 geographically dispersed farms in China between November 2007 and May 2017 (Table S2). Bacteria species were identified by biochemical tests and 16S diagnostic PCR (Oliveira, Galina & Pijoan, 2001; De la Fuente et al., 2007). All 55 G. parasuis isolates were characterised using serotyping and MLST as previously described (Wang et al., 2016; Jia et al., 2017).

Genome sequencing, assembly, and bioinformatics analysis

Isolates were cultured on tryptic soy agar or in tryptic soy broth (Oxoid, Hampshire, UK) supplemented with 10 mg/mL nicotinamide adenine dinucleotide and 5% bovine serum at 37 °C in 5% CO2 for 24 h. Total genomic DNA was extracted using the DNeasy DNA extraction kit (Axygen, Union City, CA, USA).

Among the 55 isolates, one multidrug-resistant isolate (HPS-1) and one sensitive isolate (HPS-2) from diseased swine in Guangdong were randomly selected for WGS using the PacBio RSII (Pacific Biosciences, MenloPark, CA, USA) and Illumina MiSeq (Illumina, San Diego, CA, USA) platforms as previously described (Zheng et al., 2017). The genome assemblies of HPS-1 generated in this study were deposited in GenBank under accession number CP040243. The plasmid pYL1 and transposon Tn6678 of HPS-1 were submitted to GenBank under accession number MK182379 and and MK994978, respectively. Genomic libraries of the other 53 genomes were generated and sequenced using the Illumina HiSeq 4000 system (Illumina, San Diego, CA, USA) as previously described (Soge et al., 2016). WGS data were assembled using SOAPdenovo v1.05 software (assembly statistics available in Table S3). Gene prediction was performed using GeneMarkS (Besemer, Lomsadze & Borodovsky, 2001), and a whole-genome BLAST (Altschul et al., 1990) searches (E-value ≤ 1e−5, minimal alignment length percentage ≥ 80%) against 6 databases: Kyoto Encyclopedia of Genes and Genomes (KEGG), Clusters of Orthologous Groups (COG), NCBI non-redundant protein database (NR), Swiss-Prot, Gene Ontology (GO), and TrEMBL.

Phylogenetic and clustering analyses

Two phylogenetic trees were constructed to assess the relatedness of the 55 G. parasuis strains and 39 previously published genome sequences using single-copy core orthologs and single nucleotide polymorphisms (SNPs) (Table S2). Phylogenetic inference was conducted using a maximum-likelihood optimality criterion as implemented in PhyML v3.0 (Guindon et al., 2010). The WAG amino acid substitution matrix was used for inference of the single-copy core ortholog tree, and the HKY85 nucleotide substitution model was used for inference of the SNP tree. The SNP tree was rooted using Glaesserella sp.15-184 as an outgroup. The gene contents of all 94 isolates were compared using CD-HIT (v 4.6.1) software to generate non-paralogous gene clusters (identity ≥ 0.8, ≥ 80% the length of the longest cluster).

Comparison of antimicrobial resistance and virulence genes

A whole-genome BLAST search (E-value ≤ 1e−5, minimal alignment length percentage ≥ 80%) was performed against four databases for pathogenicity and drug resistance analysis: Pathogen Host Interactions (PHI), Virulence Factors of Pathogenic Bacteria (VFDB), Carbohydrate-Active enZYmes Database (CAZy), and Integrated Antibiotic Resistance Genes Database (IARDB).

Features of the novel Tn 6678 transposon in HPS-1

Based on the results of the BLASTn search, genomic characteristics were compared among four isolates that harboured a transposon Tn6678-like structure. BLASTn searches were performed to identify genes homologous to bcr, encoding the multidrug efflux system BCR/CflA, The homologuous sequences were aligned using MUSCLE algorithm in MEGA 7.0.26 (Kumar, Stecher & Tamura, 2016) and manually adjusted, yielding 92 candidate genes. The default parameter for gap opening and gap extension were used. The phylogenetic tree was generated using MEGA 7.0.26 software using the neighbour-joining method (Kumar, Stecher & Tamura, 2016) with the Kimura 2-parameter substitution model; branch support was assessed using 1,000 bootstrap replicates.

Electrotransformation and plasmid stability test

Plasmid pYL1 harboring two antimicrobial resistance genes, blaROB−1 and aac(6′)-Ie-aph(2″)-Ia, which confer to β-lactams and aminoglycosides resistance. To determine the contributions of pYL1 to penicillin and aminoglycoside antibiotic resistance, electrotransformation experiments were performed using Staphylococcus aureus RN4220 as the recipient as previously described (Wang et al., 2015). Transformants were selected on brain-heart infusion (BHI) agar supplemented with kanamycin (25 µg/mL) for colony growth at 37 °C for 16 h. Transformation efficiency was calculated based on the ratio of transformants to the total number of viable cells. The presence of the aac(6′)-Ie-aph(2″)-Ia and bla ROB−1 genes in transformants was confirmed by PCR amplification followed by DNA sequence analysis. The primers for blaROB−1 (494 bp) were 5′-CGCTTTGCTTATGCGTCCAC-3′ (forward) and 5′-ACTTTCCACGATGTTGGCGT-3′. The primers for aac(6′)-Ie-aph(2″)-Ia (412 bp) were 5′-AGAGCCTTGGGAAGATGAAGTT-3′ (forward) and 5′-TGCCTTAACATTTGTGGCATT-3′ (reverse). The primers were designed using NCBI Primer-BLAST. The PCR conditions were as follows: initial denaturation at 95 °C for 5 min, 30 cycles of amplification (30 s at 95 °C, 30 s at 58°C, and 90 s at 72 °C), followed by extension at 72 °C for 10 min. The PCR products were purified and sequenced by Majorbio Company (Shanghai, China). The MICs of S. aureus RN4220 and five transformants were determined by Etest (Liofilchems.r.l.) according to the manufacturer’s instructions.

The stability of plasmids carrying aac(6′)-Ie-aph(2″)-Ia and blaROB−1 was determined by serial passages for 15 consecutive days at 1:1000 dilutions into fresh BHI, with or without antibiotic (kanamycin) pressure. Serially diluted cultures were spread on BHI agar plates with or without kanamycin (8 µg/mL), and the resistance retention rate was determined by randomly picking at least 50 colonies from the BHI plates, spotting them onto new BHI plates with kanamycin (8 µg/mL), and calculating the ratio of resistant to total colonies. Both the resistant and susceptible colonies from the plates were randomly picked and subjected to PCR for detection of blaROB−1 and aac(6′)-Ie-aph(2″)-Ia.

Results

G. parasuis core and unique genes

Compilation of the 94 genomes covering all serovars and disease- and non-disease-causing backgrounds from nine geographic locations (Table S4) demonstrated expansion of the pan-genome, whereas the number of core genes remained relatively stable with the addition of new strains (Fig. 1A). This result suggests the presence of an open pan-genome experiencing frequent evolutionary changes through gene gains and losses or lateral gene transfer. The size of the pan-genome was 5,243 genes, including ∼3.34% core genes shared among the 94 isolates mainly from China (Fig. 1B). Meanwhile, accessory genomes occupied a large fraction (85.13–91.74%) of the G. parasuis gene content compared with the core genomes and the number of unique genes ranged from 0 to 103 indicating that 0–4.6% of the genome consists of strain-specific accessory genes (Table S2).

Figure 1 Analysis of the core and pan-genome of G. parasuis isolates.

(A) Core and pan-genomic calculations in G. parasuis isolates. Each green point represents the number of genes conserved between genomes. All of the points are plotted as a function of the strain number (x). The deduced pan-genome size: P(x) = 2483.54x0.18 − 461.72. The height of the curve continues to increase because the pan-genome of G. parasuis is open. (B) Genes missing or present in G. parasuis isolates. The heat map illustrates the distribution of core and accessory genes across the G. parasuis strains. The columns represent G. parasuis isolates. The rows represent genes. The red and black regions represent the presence or absence of genes in a particular genome, respectively. The black regions indicate features missing in that strain but present in one or more of the other G. parasuis strains. (C) The distribution of all, core, and specific genes according to the COG classification. The y-axis indicates the percentage of genes in various COG categories. A, RNAprocessing and modification. C, Energy production and conversion. D, Cell cycle control, cell division, chromosome partitioning. E:Amino acid transport and metabolism. F, Nucleotide transport and metabolism. G: Carbohydrate transport and metabolism. H, Coenzyme transport and metabolism. I, Lipid transport and metabolism. J, Translation, ribosomal structure and biogenesis. K, Transcription. L, Replication, recombination and repair. M, Cell wall/membrane/envelope biogenesis. N, Cell motility. O, Posttranslational modification, protein turnover, chaperones. P, Inorganic ion transport and metabolism. Q, Secondary metabolites biosynthesis, transport and catabolism. R, General function prediction only. S, Function unknown. T, Signal transduction mechanisms. U, Intracellular trafficking, secretion, and vesicular transport. V, Defense mechanisms.

Clusters of Orthologous Groups classification indicated that core genes were significantly enriched in defense mechanisms and inorganic ion transport and metabolism, whereas unique genes were significantly enriched in unknown function, nucleotide transport and metabolism, and carbohydrate transport and metabolism (Fig. 1C).

Phylogenetic analysis of G. parasuis isolates

A phylogenetic tree based on single-copy core genes of our isolates and reference isolates resolved two well-supported lineages, lineages I and II, exhibiting association with country, serotypes, and MLST types (Fig. 2). Lineages I and II comprised eight and two countries, respectively. Serovars 5, 12, and 14 were identified predominantly in lineage I, while serovars 2 and 10 were mostly found in lineage II. For serovars 3, 8, 9, and 11, the numbers of isolates were too low to draw conclusions about phylogenetic patterns. The remainder of the serovars were found in both clades.

Figure 2 Maximum-likelihood phylogeny of 94 Glaesserella parasuis isolates based on 93 single-copy core genes.

The tree was constructed with MEGA 7 with 1,000 bootstrap replicates. The annotation rings surrounding the tree, from outside to inside, depict (1) geographic region, (2) year of sample collection, (3) site of sample, and (4) serotype. The different colors of the branches represent lineages, lineage in pink and lineage in green.

MLST analysis assigned the 39 isolates in GenBank to 20 different STs, including six new STs, with 13 isolates not determined. The 55 isolates obtained in our study belonged to 49 different STs, including 39 new STs (Table S2). Most strains of the same STs formed single clades (Fig. 2). The SNP-based tree with and without an outgroup (Figs. S1 and S2) was consistent with the phylogenetic analysis based on single-copy core orthologs. The number of whole-genome SNP differences among the 94 isolates ranged from 8,603 to 8,730.

Biological features of G. parasuis isolates

Variation in virulence and stress resistance genes was observed among G. parasuis lineages and subgroups (Fig. 3). All 94 G. parasuis isolates harboured more than five types of pathogenic factors. The virulence factors gigP, malQ, and gmhA were carried by all the tested G. parasuis isolates. Moreover, other virulence factors including the rfa cluster, encoding enzymes for lipopolysaccharide (LPS) core biosynthesis, and galU and galE, resulting in impaired biofilm formation, were universally present in the G. parasuis isolates.

Figure 3 Virulence and resistance profiles across the phylogeny of the 94 G. parasuis isolates.

Cluster analysis based on single-copy core orthologs. Pattern of gene presence (colored line) or absence (white).

The main ARGs associated with resistance in G. parasuis, including the β-lactam-resistant gene bla ROB−1, tetracycline resistance genes tetB, aminoglycoside resistance genes aph(3′)-Ib and aac(6′)-Ie-aph(2″)-Ia, fluoroquinolone resistance gene norA, chloramphenicol resistance genes catIII and floR, sulfonamide resistance gene sul2 were discovered (Fig. 3). Among all of these genes, the genes sul2 and aph(3′)-Ib, and β-lactam-resistant genes pbp1a and pbp3a were universally present in the G. parasuis isolates (Fig. 3). Three different serotype isolates (H82, H92, and H313) obtained from different sites in different years that clustered closely in one branch all harboured the lincosamide antibiotic resistance factor lunC (Fig. 3). Moreover, 91.5% of the isolates had bcr, 90.42% of the isolates had bacA, 100% of the isolates had ksgA, but five isolates had norA.

Genomic features of G. parasuis HPS-1

Following sequencing and assembly, a 2,326,414-bp chromosome with an average G+C content of 40.03%, and a 7,777-bp small plasmid sequence (pYL1) with an average G+C content of 33.32% were identified in strain HPS-1 (Fig. S3 and Fig. 4). HPS-1 exhibited a novel ST (ST328) with undescribed MLST alleles or previously unreported allelic combinations. This ST328 genome harbored resistance genes against several types of antibiotics, including sulfonamides (sul2), aminoglycosides (aph(3′)-Ib, aac(6′)-Ie-aph(2″)-Ia), and β-lactam (blaROB−1) (Table S1). Further, this genome contained efflux pump-related genes that confer resistance to sulfonamides (bcr) and multidrug resistance (acrB).

Figure 4 Schematic map of plasmid pYL1.

The circles show, from outside to inside: first and second, putative open reading frames, the positions and orientations of the genes; third, G+C content (deviation from the average); and, fourth, G+C skew (green, +; purple, −).

We also identified a novel transposon in the ST328 isolate, designated Tn6678 in the Tn Number Registry (https://transposon.lstmed.ac.uk/). This transposon harbours two 966-bp IS110 family transposases at both ends, two toxin genes pilT and phd, two genes associated with the two-component signal transduction system cpxA and cpxR, one efflux pump-associated gene bcr, and four genes encoding hypothetical proteins with unknown function (Fig. 5). Genome analysis revealed that Tn6678 was inserted between the molybdopterin molybdotransferase MoeA encoded by moeA and 3-isopropylmalate dehydratase large subunit encoded by leuC. A LacI family transcriptional regulator and a bifunctional tRNA (5-methylaminomethyl-2-thiouridine)(34)-methyltransferase MnmD/FAD-dependent 5-carboxymethylaminomethyl-2-thiouridine (34) oxidoreductase MnmC flanked the transposon to the right and left, respectively.

Figure 5 Organization of the G. parasuis HPS-1 Tn6678 transposon and comparison with the similar structure.

ORFs are shown as arrows, indicating the transcription direction, and the colors of the arrows represent different fragments. Gene color code: transposase, purple; toxin genes (pilT and phd), yellow; resistance genes (cpxA, cpxR and bcr), blue; proteins with other or unknown functions, gray. Homologous gene clusters in different isolates are shaded in gray (>97%).

Through BLASTN searches, highly conserved homologous sequences to Tn6678 (>97% nucleotide sequence similarity) were identified in four G. parasuis strains [29755 (GenBank accession number CP021644, USA), SH0165 (CP001321, China), ZJ0906 (CP005384, China), and str. Nagasaki (NZ_APBT00000000, Japan)]. The only differences in these five chromosomes were in the transposases, but transposon Tn6678 had two complete inverted repeats of IS110 transposases flanked by 32-bp inverted repeats of ISNme5 at both ends (Fig. 5), suggesting mobility potential. The bcr-containing Tn6678 also contained an antibiotic resistance gene cassette, suggesting its potential to transfer antibiotic resistance genes.

BLASTn searches for the bcr gene returned a large set of divergently related sequences using default parameters. These sequences were annotated as bicyclomycin/multidrug efflux system, Bcr/CflA family drug resistance efflux transporter, Bcr/CflA family multidrug efflux major facilitator superfamily (MFS) transporter or drug resistance transporter, and Bcr/CflA subfamily. Phylograms revealed that the bcr gene in HPS-1 was most closely related to homologs identified in other members of the Pasteurellaceae, particularly G. parasuis, Actinobacillus indolicus, Bibersteinia trehalosi, Actinobacillus (A. pleuropneumoniae, A. suis, A. equuli, A. lignieresii, A. indolicus, and A. porcitonsillarum), and Mannheimia (M. haemolytica and M. varigena), all of which are known causative agents of upper respiratory tract infections (Fig. 6).

Figure 6 Neighbor-joining phylogenetic tree based on bcr gene sequences obtained from the current study and downloaded from NCBI.

The tree was constructed using MEGA 7 with 1,000 bootstrap replicates. The different colors of the branches represent lineages. The G. parasuis HPS-1 is indicated by a solid circle.

The neighbour-joining phylogenetic tree using 92 bcr genes selected from the BLASTn searches clearly demonstrated two distinctive clades. The first clade contained bcr genes of Hemophilus influenzae, which colonizes humans, and other Haemophilus species that colonize non-human animals. Members of the second clade were divided into four apparent subclades, including G. parasuis, B. trehalosi, Actinobacillus spp., and Mannheimia spp. Except for G. parasuis, the chromosomally encoded Bcr/CflA from G. parasuis HPS-1 most closely clustered with that found in A. indolicus. The phylogenetic tree indicated a divergent evolutionary pattern between animal-origin Pasteurellaceae bacteria. The bcr gene tree is consistent with the organismal phylogeny, suggesting that horizontal gene transfer does not play an important role in the evolution of bcr-mediated resistance.

General features and electrotransformation of the plasmid pYL1

The plasmid pYL1 identified in HPS-1 contained seven ORFs with an average length of 912 bp, with one encoded protein of undetermined function (Fig. 4), and two antimicrobial resistance genes, blaROB−1 and aac(6′)-Ie-aph(2″)-Ia. Four ORFs were identified to encode a 3′-truncated transposase protein ISApl1 (30 amino acids), a Rep-like protein (444 amino acids) involved in plasmid replication, and two Mob proteins, MobC (144 amino acids) and MobA (541 amino acids), associated with plasmid mobilization (Fig. 4). Except for resistance genes, pYL1 had the same backbone and genetic structure and showed 100% nucleotide identity to four previously-identified plasmids, pFZ51, pFS39, pHN61, and pHB0503 (Table S5) (Kang et al., 2009; Chen et al., 2010; Yang et al., 2013). In contrast, the resistance genes and flanking regions in pYL1 exhibited as little as 58% sequence identity to the other four plasmids (Fig. 7).

Figure 7 Comparison of the genetic structures of pHN61, pFS39, pYL1, pFZ51 and pHB0503.

The accession numbers and origins of these plasmids are displayed on the left side. Arrows represent putative open reading frames, the positions and orientations of the genes. Blue arrows indicate Rep-like protein involved in plasmid replication. Green arrows indicate hypothetical protein. Regions with more than 98% nucleotide sequence identity are shaded yellow.

Transformation of pYL1 into S. aureus RN4220 was achieved at a frequency of 10−9 cells per recipient cell by electroporation, confirming that pYL1 is a mobilizable plasmid with active mobilization genes. The transformants had increased MICs for oxacillin, gentamicin, amikacin, kanamycin, and streptomycin as compared with those of the parental strain (0.047 to >256 mg/L, 0.094 to 1.5 mg/L, 0.38 to 16 mg/L, 0.38 to 32 mg/L, and <0.25 to 32 mg/L, respectively). This finding indicated that plasmid pYL1 carrying blaROB−1 and aac(6′)-Ie-aph(2″)-Ia contributed to the penicillin resistance and aminoglycoside antibiotic resistance in S. aureus RN4220 transformants. Furthermore, the plasmid showed low stability in S. aureus without antibiotic pressure, as only 52.5%, 30.48%, and 2.68% of transformants maintained the kanamycin resistance after five, six, and seven subcultures, respectively. However, the plasmid can be conserved in S. aureus cultured with kanamycin, as 100% of the colonies remained resistant to kanamycin after 10 subcultures, as confirmed by PCR mapping.

Discussion

In the current study, we observed an open pan-genome. Similar result that the size of pan-genome was 7,431 genes including 1,049 core genes has been reported (Howell et al., 2014). This suggested that the G. parasuis pan-genome is vast, and unique genes can be continuously be identified upon sequencing more G. parasuis genomes. However, the isolates in this study with ∼3.34% core genes, primarily isolated from China, displayed further diversity and higher variability than isolates with only ∼14% core genes, primarily obtained from the UK (Howell et al., 2014). Besides, we identified 54 new STs enriching the G. parasuis MLST databases and highlight the wide distribution of G. parasuis strains. Although most strains of the same STs formed single clades, there was no definitive association between ST and serotype (Fig. 2), consistent with previous studies (Olvera, Cerda-Cuellar & Aragon, 2006; Wang et al., 2016).

The pattern of the phylogenetic tree based on single-copy core genes was different from the population grouping predicted via MLST, which showed six main subgroups (Wang et al., 2016). Both phylogenetic lineages contain both Asian and North American isolates, in agreement with previous phylogenetic analyses (Howell et al., 2014; Wang et al., 2016; Dickerman, Bandara & Inzana, 2020) and supporting the hypothesis of frequent migration of isolates between geographic regions.

Five types of pathogenic factors gigP, malQ, gmhA, rfa and gal cluster were universally carried by G. parasuis isolates in this study. The rfaF gene has been linked to serum resistance, adhesion, and invasion (Zhang et al., 2013); galU plays a role in autoagglutination and biofilm formation, and galE appears to affect biofilm production indirectly in G. parasuis (Zou et al., 2013). Serum resistance may play a role in the virulence of G. parasuis (Cerda-Cuellar & Aragon, 2008). However, lsgB, previously associated with G. parasuis virulence potential, was predominant in six isolates (29755 and HPS9 from the USA, Nagasaki from Japan, and KL0318, SH0104, and SH0165 from China), in line with potentially virulent strains isolated from the nasal cavities of healthy pigs (Amano et al., 1996; Brockmeier et al., 2013).

The blaROB−1, sul2, aph(3′)-Ib, tetB, tetD, aac(6′)-Ie-aph(2″)-Ia, catIII, and floR genes have previously been identified in G. parasuis (Zhao et al., 2018). In the current study, we identified all of genes mentioned above. This is the first report of genes tetA, tetH and tetR genes in G. parasuis isolates and needs further study. Tetracycline resistance genes are often associated with conjugative and mobile genetic elements enabling horizontal transfer (Lancashire et al., 2005; Zhao et al., 2018). Moreover, this is the first report describing the presence of the bcr, bacA, ksgA and norA genes in G. parasuis, to the best of our knowledge. All of these benefits from the application of whole genome sequencing method. Three isolates clustered closely in one branch all harboured lunC gene, contained in the ISSag10 sequence of all three isolates. The lunC gene was only identified in plasmid pHN61 of G. parasuis (Chen et al., 2010). The results suggested that the resistance of these three strains to lincomycin may be mediated by the plasmid carrying lunC gene.

This is also the first report describing the transoson Tn6678 containing toxin genes pilT and phd, drug resistance genes cpxA and cpxR, and an efflux pump gene bcr. Association between the Cpx system and bacterial antimicrobial resistance has been reported in Escherichia coli, Salmonella enterica, Klebsiella pneumoniae, and G. parasuis (Hu et al., 2011; Srinivasan et al., 2012; Audrain et al., 2013; Kurabayashi et al., 2014; Cao et al., 2018). CpxR plays essential roles in mediating macrolide (i.e., erythromycin) resistance (Cao et al., 2018). The Bcr/CflA efflux system was identified as a group of antiporters that confer resistance to chloramphenicol, florfenicol, and bicyclomycin by actively transporting these compounds out of the cell (Marklevitz & Harris, 2016). The transposon Tn6678 had two complete inverted repeats of IS110 transposases flanked by 32-bp inverted repeats of ISNme5 at both ends suggesting mobility potential and its potential to transfer antibiotic resistance genes. In G. parasuis, only the efflux pump AcrB, belonging to the resistance-nodulation division (RND) family, has been analysed to date. Efflux pump AcrB may play a role in multidrug resistance, and the acrAB gene cluster could affect the efflux of macrolides in G. parasuis (Feng et al., 2014). However, this is the first description of the efflux pump Bcr/CflA in G. parasuis, belonging to the MFS. This efflux pump, encoded by bcr, harbored on a transposon indicated its potential transferability.

To date, two β-lactam resistance genes (bla ROB−1 and blaTEM ) have been reported in G. parasuis (By (Guo et al., 2012). A β-lactam resistance plasmid, pB1000, harbouring blaROB−1 was previously detected in G. parasuis clinical strains isolated from Glässer’s disease lesions (San et al., 2007). The plasmid pYL1 harboured two antimicrobial resistance genes, blaROB−1 and aac(6′)-Ie-aph(2″)-Ia. The ROB-1 of plasmid pYL1 had a typical size of 305 bp, in line with functionally active members of the ROB-1 family from different plasmids in Pasteurellaceae species. AAC(6′)-Ie-APH(2′)-Ia, the most important aminoglycoside-resistance enzyme in gram-positive bacteria conferring resistance to almost all known aminoglycoside antibiotics in clinical use, also had a typical size of 479 amino acids in this family (Rouch et al., 1987). Although aac(6′)-Ib-cr is considered the most prominent aminoglycoside-resistance gene in G. parasuis (Doi & Arakawa, 2007; San et al., 2007), the bifunctional aminoglycoside-resistance enzyme AAC(6′)-Ie-APH(2′)-Ia in plasmids is also reported in GenBank for G. parasuis strains. Comparing with other four previously-identified plasmids which have similar structure with pYL1 suggested more rapid evolution among the resistance-associated components of these small plasmids. The transposase gene of ISApl1 in pYL1 had an internal deletion of 659 bp, but intact 3′ and 5′ ends. The truncated ISApl1 linked with blaROB−1 suggested that ISApl1 played a key role in transposition of blaROB−1, facilitating the horizontal transfer of β-lactam and aminoglycoside resistance among G. parasuis isolates. These results are consistent with a previous study presenting evidence for spread of β-lactam resistance (Yang et al., 2013). A similar occurrence was also identified in A. porcitonsillarum or G. parasuis plasmids pFJS5863, pQY431, and pFS39, suggesting a more widespread role and highlighting that the function of ISApl1 requires further investigation.

Conclusions

In summary, our results shed new light on the importance of genomic variations, especially transposon-related and/or plasmid-related gene variations, in the evolution of G. parasuis. This comparative analysis identified potentially novel virulence factors (gigP, malQ, and gmhA) and drug resistance genes (norA, bacA, ksgA, and bcr) in G. parasuis. Resistance determinants (sul2, aph(3′)-Ib, norA, bacA, ksgA, and bcr) were widespread across isolates, regardless of serovar, isolation source, or geographical location. Future research focused on a larger sample of G. parasuis isolates worldwide will further increase understanding of the rapid development of antibiotic resistance associated with mobile genetic elements in this important animal pathogen.

Supplemental Information

Table S1 Characteristics of G. parasuis strain HPS-1, including antimicrobial resistance profile and presence of resistance genes

MIC: minimum inhibitory concentration; R: resistant; S: susceptible. a Interpreted according to Clinical and Laboratory Standards Institution (CLSI) guidelines for P. aeruginosa [Clinical and Laboratory Standards Institute. Performance standards for antimicrobial susceptibility testing; twenty-fifth informational supplement. Wayne, PA: CLSI; 2015 Document M100-S25.].

Click here for additional data file.

Table S2 Sequenced G. parasuis genomes involved in this study

N.A. represents unknown information; ST: sequence type; UT: untypeable.

Click here for additional data file.

Table S3 Assembly results for 55 G. parasuis genomes sequenced in the present study

Click here for additional data file.

Table S4 Orthologous clusters observed in the G. parasuis pan-genome

Click here for additional data file.

Table S5 Characteristics of plasmids compared in this study

Click here for additional data file.

Figure S1 Maximum-likelihood phylogeny of 94 Glaesserella parasuis isolates based on whole-genome single nucleotide polymorphisms (SNPs)

The tree was constructed with MEGA 7 using a maximum-likelihood optimality criterion as implemented in PhyML v3.0 with 1, 000 bootstrap replicates. The annotation rings surrounding the tree, from inside to outside, depict (1) serotype, (2) host, (3) geographic region and (4) year of sample collection. The branch colors denote two major lineages, lineage I (pink) and lineage II (green).

Click here for additional data file.

Figure S2 Maximum likelihood phylogeny of 94 G. parasuis isolates constructed using PhyML to analyze the whole-genome SNP dataset.

Glaesserella sp. 15–184 was chosen as an outgroup. The branch colors denote two major lineages, lineage I (pink) and lineage II (green).

Click here for additional data file.

Figure S3 Circular map of Glaesserella parasuis strain HPS-1

From inside to outside, the first circle represents the genome size of HPS-1; the second circle represents the GC skew; the third circle represents the GC content; the fourth circle and the seventh circle represent the COG (cluster of orthologous groups) designation of each coding sequence (CDS); the fifth and sixth circles represent the position of CDS, tRNA, and rRNA on the genome.

Click here for additional data file.

Supplemental Information 1 Comparison of antimicrobial resistance and virulence genes

Click here for additional data file.

Supplemental Information 2 Abbreviation table

Click here for additional data file.

Supplemental Information 3 The sequences of H43

Click here for additional data file.

Supplemental Information 4 The sequences of H27

Click here for additional data file.

Supplemental Information 5 The sequences of H26

Click here for additional data file.

Supplemental Information 6 The sequences of H40

Click here for additional data file.

Supplemental Information 7 The sequences of H46

Click here for additional data file.

Supplemental Information 8 The sequences of H45

Click here for additional data file.

Supplemental Information 9 The sequences of H52

Click here for additional data file.

Supplemental Information 10 The sequences of H100

Click here for additional data file.

Supplemental Information 11 The sequences of H61

Click here for additional data file.

Supplemental Information 12 The sequences of H49

Click here for additional data file.

Supplemental Information 13 The sequences of H60

Click here for additional data file.

Supplemental Information 14 The sequences of H33

Click here for additional data file.

Supplemental Information 15 The sequences of H64

Click here for additional data file.

Supplemental Information 16 The sequences of H68

Click here for additional data file.

Supplemental Information 17 The sequences of H74

Click here for additional data file.

Supplemental Information 18 The sequences of H82

Click here for additional data file.

Supplemental Information 19 The sequences of H78

Click here for additional data file.

Supplemental Information 20 The sequences of H80

Click here for additional data file.

Supplemental Information 21 The sequences of H92

Click here for additional data file.

Supplemental Information 22 The sequences of H87

Click here for additional data file.

Supplemental Information 23 The sequences of H90

Click here for additional data file.

Supplemental Information 24 The sequences of H106

Click here for additional data file.

Supplemental Information 25 The sequences of H140

Click here for additional data file.

Supplemental Information 26 The sequences of H105

Click here for additional data file.

Supplemental Information 26 The sequences of H110

Click here for additional data file.

Supplemental Information 27 The sequences of H115

Click here for additional data file.

Supplemental Information 28 The sequences of H112

Click here for additional data file.

Supplemental Information 29 The sequences of H143

Click here for additional data file.

Supplemental Information 30 The sequences of H134

Click here for additional data file.

Supplemental Information 31 The sequences of H178

Click here for additional data file.

Supplemental Information 32 The sequences of H159

Click here for additional data file.

Supplemental Information 33 The sequences of H190

Click here for additional data file.

Supplemental Information 34 The sequences of H160

Click here for additional data file.

Supplemental Information 35 The sequences of H164

Click here for additional data file.

Supplemental Information 36 The sequences of H191

Click here for additional data file.

Supplemental Information 37 The sequences of H197

Click here for additional data file.

Supplemental Information 38 The sequences of H199

Click here for additional data file.

Supplemental Information 39 The sequences of H222

Click here for additional data file.

Supplemental Information 40 The sequences of H157

Click here for additional data file.

Supplemental Information 41 The sequences of H201

Click here for additional data file.

Supplemental Information 42 The sequences of H233

Click here for additional data file.

Supplemental Information 43 The sequences of H223

Click here for additional data file.

Supplemental Information 44 The sequences of H259

Click here for additional data file.

Supplemental Information 45 The sequences of H257

Click here for additional data file.

Supplemental Information 46 The sequences of H25

Click here for additional data file.

Supplemental Information 47 The sequences of H299

Click here for additional data file.

Supplemental Information 48 The sequences of H19

Click here for additional data file.

Supplemental Information 49 The sequences of H285

Click here for additional data file.

Supplemental Information 50 The sequences of H275

Click here for additional data file.

Supplemental Information 51 The sequences of H292

Click here for additional data file.

Supplemental Information 52 The sequences of H263

Click here for additional data file.

Supplemental Information 53 The sequences of H313

Click here for additional data file.

Supplemental Information 54 The sequences of H312

Click here for additional data file.

Supplemental Information 55 The sequences of HPS-2

Click here for additional data file.

We thank members of our laboratories for fruitful discussions.

Additional Information and Declarations

Competing Interests

Author Contributions

DNA Deposition

The authors declare there are no competing interests.

Xiulin Wan conceived and designed the experiments, performed the experiments, analyzed the data, prepared figures and/or tables, authored or reviewed drafts of the paper, and approved the final draft.

Xinhui Li and Todd Osmundson authored or reviewed drafts of the paper, and approved the final draft.

Chunling Li conceived and designed the experiments, authored or reviewed drafts of the paper, and approved the final draft.

He Yan conceived and designed the experiments, analyzed the data, authored or reviewed drafts of the paper, and approved the final draft.

The following information was supplied regarding the deposition of DNA sequences:

The genome assemblies of HPS-1 are available at GenBank: CP040243. The plasmid pYL1 and transposon Tn6678 of HPS-1 are available at GenBank: MK182379 and MK994978.

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
