# Peer review of "Whole-genome sequence analyses of Glaesserella parasuis isolates reveals extensive genomic variation and diverse antibiotic resistance determinants"

_PeerJ, doi:10.7717/peerj.9293_

## Round 0.1 · original submission · Major Revisions

Dear Dr. Wan and colleagues:

Thanks for submitting your manuscript to PeerJ. I have now received three independent reviews of your work, and as you will see, the reviewers raised some concerns about the research. Despite this, these reviewers are optimistic about your work and the potential impact it will lend to research on genomic variation and factors underpinning antibiotic resistance in Glaesserella parasuis. Thus, I encourage you to revise your manuscript, accordingly, taking into account all of the concerns raised by the reviewers.

While the concerns of the reviewers are relatively minor, this is a major revision to ensure that the original reviewers have a chance to evaluate your responses to their concerns.

Please make sure all of the missing information (including methodologies) pointed out by the reviewers is provided in your revision. Your study needs to be entirely repeatable. Please restructure the Introduction to be clearer as far as providing enough background to understanding the study and its relevance. Ensure all figures are legible.

I look forward to seeing your revision, and thanks again for submitting your work to PeerJ.

Good luck with your revision,

-joe

Reviewer 1 ·

Basic reporting

The language is clear and unambiguous.
Although Table 1 is important, it is too long for the reader. Unless it could be shown in a figure, the table can be shown in the supplemental files.

Experimental design

The substitution models were not mentioned in the section of methods.

Validity of the findings

1. No bootstrap values in Figure 2 and Figure 6.
2. The quality the NGS results were missing, such as N50 and sequencing depth.

Additional comments

In the manuscript, Wan et al. has performed a whole genome sequence analysis of Glaesserella parasuis from different isolate. They also described the genomic features of those isolate, and found that a new transposon and plasmid which coffered resistance. They also performed phylogetics analysis to see the cluster of isolate with different background, and therefore showed the diversity of the accessory genes. In general, this manuscript is well designed and clearly written, although some minor revisions should be made before acceptance.

Reviewer 2 ·

Basic reporting

This study provided a valuable information of the whole-genome sequencing analysis of Glaesserella parasuis isolates.

Experimental design

No comment.

Validity of the findings

The results provided important information for pathogenesis and treatment of G. parasuis in the future.

Additional comments

In this manuscript, the authors provided a valuable information of the whole-genome sequencing analysis of Glaesserella parasuis isolates. I believe this paper article provided important information for pathogenesis and treatment of G. parasuis in the future.

Major comments:
1. This study talk about the antibiotic resistance genes (ARGs) and virulence factors, so authors should be give more background of ARGs and virulence factors in introduction part.
2. Some of figures are too blurry and difficult to read, should be increase the resolution. For example, Fig 2, Fig 3, and Fig 6. Also Fig S1 and Fig S2.
3. The format of reference is not Peer J, authors should be follow up and updated.

Reviewer 3 ·

Basic reporting

The manuscript describes the WGS analysis of 55 genomes from isolates of G. parasuis, a main pathogen to swine industry. Although there have been other reports on WGS on this particular bacteria, it is important to characterize more isolates where the geographical information together with the pathogenicity to increase the knowledge on this dual bacteria that can be either commensal or pathogenic for the herd.
The phylogenetic analysis plus the antimicrobial and virulence markers are described herein. Moreover, the characterization of a potential new plasmid is done. The background is confusing since many times it’s not clear if the authors are explaining their results or the ones obtained previously (an example would be line 53). There is a clear mistake about the number of strains analyzed in reference 10. Many times the sentences are not connected and it’s difficult to understand the reason why the authors are mentioning the fact. More connection is needed to be flow.

Experimental design

The M&M section is brief and with scarce information, many times including just the references to explain the method used, so it is difficult to follow what the authors have actually done to obtain results (examples are 102-105). (The methods section from the abstract does not report any method either and repeats a background sentence).
The phylogenetic analyses were done with neighbor joining method, however it is not mentioned whether the estimation of the best method was assessed.
I don’t really see the need for including a brief paragraph as supplementary material to include a main method to discover antimicrobial genes (line 123).
Sometimes the methods are mentioned but the detailed analysis remains obscure. Examples of this would be “were aligned… and adjusted manually” (line 128); probably due to communication issues. And “…confirmed by PCR amplification…” primers? Condition? Reference? (line138)
Many times, the reason behind the selection of the method is not stated (the gap open and extent was modified to -400 Line 129).

Validity of the findings

The results & discussion section is not easy to follow either, for multiple reasons. One of them is the lack of information in M&M which makes difficult to understand how the results were obtained. Again, sometimes reported results are actually coming from the literature but is not clear and is mixed with results from this manuscript (example lines 156-161). Some sentences are not clear (line 163) or hypothesis are not well supported by the results (ex. line 165-167). Figure 1C is difficult to read since the letter are all mixed up in the caption. Sometimes things are selected without mentioning the scientific reason of the selection (line 271).

Additional comments

Some other important things that need to be addressed are:
Were the associations tested with some statistical tests? (lines 178-184 and 190-192).
What is the biological meaning that the SNP and the single-copy core gene trees are consistent?
What about the associations among resistances and geographic location? And antibiotic usage in the particular location?
The importance of bcr gene and the explanation why the analysis was deeper for this particular gene is not clearly mentioned.
Explain why do you suggest “evolution among these small plasmids” (line 299)
“PCR mapping” so you mean PCR amplification? (line 318)
Abbreviation table with the genes mentioned is needed, many abbreviations are not introduced in the text (examples MFS, )
Figures are not clear enough, especially fig 1, 3 and 7 (what has been shaded in grey?).
There is some excess of reporting figures without the clear information that they provide.

---

## Round 0.2 · accepted · Accept

Dear Dr. Wan and colleagues:

Thanks for revising your manuscript based on the concerns raised by the reviewers. I now believe that your manuscript is suitable for publication. Congratulations! I look forward to seeing this work in print, and I anticipate it being an important resource for groups studying genomic variation and factors underpinning antibiotic resistance in Glaesserella parasuis. Thanks again for choosing PeerJ to publish such important work.

Best,

-joe

Reviewer 1 ·

Basic reporting

The language is clear and unambiguous.

Experimental design

The experimental design is well presented.

Validity of the findings

The findings is sound and clearly stated.

Additional comments

The manuscript is greatly improved from its previous version, and therefore is suitable for acceptance.

Reviewer 2 ·

Basic reporting

I think the R1 version is much improved. The reviewer suggests to accept the manuscript.

Experimental design

no comment

Validity of the findings

no comment

Additional comments

I think the R1 version is much improved. The reviewer suggests to accept the manuscript.

Reviewer 3 ·

Basic reporting

This study provided relevant information of different Glaesserella parasuis isolates through WGS analysis.

Experimental design

Most of the points raised by this reviewer have been addressed

Validity of the findings

Meaningful findings are provided and clearly stated. Conclusions are well stated.
I recommend the authors to use updated MegaX version next time.

Additional comments

The manuscript has noticeably improved with this revision.